# MicroRNA-135b-5p Is a Pathologic Biomarker in the Endothelial Cells of Arteriovenous Malformations

**DOI:** 10.3390/ijms25094888

**Published:** 2024-04-30

**Authors:** Joon Seok Lee, Gyeonghwa Kim, Jong Ho Lee, Jeong Yeop Ryu, Eun Jung Oh, Hyun Mi Kim, Suin Kwak, Keun Hur, Ho Yun Chung

**Affiliations:** 1Department of Plastic and Reconstructive Surgery, School of Medicine, Kyungpook National University, Daegu 41944, Republic of Korea; leejspo@knu.ac.kr (J.S.L.); clerk0823@naver.com (J.H.L.); rjyflying@naver.com (J.Y.R.); fullrest74@hanmail.net (E.J.O.); sarang7939@naver.com (H.M.K.); suin8349@naver.com (S.K.); 2Department of Biochemistry and Cell Biology, School of Medicine, Kyungpook National University, Daegu 41199, Republic of Korea; med.aurora1106@gmail.com; 3Cell and Matrix Research Institute, School of Medicine, Kyungpook National University, Daegu 41944, Republic of Korea

**Keywords:** arteriovenous malformation, microRNA-135b-5p, endothelial cells

## Abstract

Arteriovenous malformations (AVMs) are congenital vascular anomalies with a poor prognosis. AVMs are considered intractable diseases, as there is no established approach for early diagnosis and treatment. Therefore, this study aimed to provide new evidence by analyzing microRNAs (miRNAs) associated with AVM. We present fundamental evidence for the early diagnosis and treatment of AVM by analyzing miRNAs in the endothelial cells of AVMs. This study performed sequencing and validation of miRNAs in endothelial cells from normal and AVM tissues. Five upregulated and two downregulated miRNAs were subsequently analyzed under hypoxia and vascular endothelial growth factor (VEGF) treatment by one-way analysis of variance (ANOVA). Under hypoxic conditions, miR-135b-5p was significantly upregulated in the AVM compared to that under normal conditions, corresponding to increased endothelial activity (*p*-value = 0.0238). VEGF treatment showed no significant increase in miR-135b-5p under normal conditions, however, a surge in AVM was observed. Under both hypoxia and VEGF treatment, comparison indicated a downregulation of miR-135b-5p in AVM. Therefore, miR-135b-5p was assumed to affect the pathophysiological process of AVM and might play a vital role as a potential biomarker of AVMs for application related to diagnosis and treatment.

## 1. Introduction 

Each type of vascular anomaly manifests various surface characteristics, appearing flat or raised, and exhibits various colors, such as blue, pink, purple, or red. According to their clinical features and history, vascular anomalies are broadly categorized into vascular tumors and malformations, with vascular malformations arising from errors in the vascular development process [1].

They are classified into capillary malformations, venous malformations, arteriovenous malformations, and lymphatic malformations, depending on the consistent vessels, each exhibiting distinct characteristics [2]. Among these, arteriovenous malformations (AVMs) have the most unfavorable prognosis, secondary to shunts between arteries and veins.

This rare vascular disease results from a direct connection of a feeding artery to a draining vein, with high-flow velocity and without an intervening capillary network. It can develop in any part of the body and may be small and asymptomatic, however, over time, it can progress to a potentially life-threatening severe impairment. The pathophysiology is closely related to angiogenesis, where high-pressure arterial blood is shunted directly into low-pressure veins that lead to vessel dilation and recruitment of new blood vessels. This continuous stimulation results in high-flow circulation, creating a relatively hypoxic condition in the surrounding tissue near the AVM.

The vascular endothelium is a multifunctional organ that undergoes adaption for maintaining homeostasis. Endothelial cells (ECs) play a vital role in metabolic activation through paracrine, endocrine, and autocrine functions along with vascular homeostasis under physiological conditions [3,4,5,6,7,8,9]. All processes involving ECs require the precise synchronization of molecular and cellular events triggered by stimulatory and inhibitory signals, ultimately leading to a physiological regulation. Consequently, highly dynamic and dose-sensitive signaling complexes become prime candidates for the microRNA (miRNA) posttranscriptional-mediated regulation of gene expression [10,11,12,13]. Similarly, in numerous situations, ECs in the blood vessels are involved in molecular and cellular process regulation in terms of permeability, leukocyte adherence, proliferation, thrombosis, and more. Furthermore, AVMs undergo vascular remodeling, which might be influenced by vascular endothelial growth factor (VEGF) levels. VEGF is an effective inducer of angiogenesis and was initially characterized as a crucial growth factor for vascular ECs. Its upregulation is observed in numerous tumors, and its role in facilitating tumor angiogenesis has been clearly established [14,15]. As VEGF is a critical regulator of vascular function and has a potential impact on AVM pathogenesis, its therapeutic effects are noteworthy [16,17,18].

Tissue oxygenation depends on the balance between oxygen supply, delivered by the vasculature, and the metabolic demand of tissues. Hypoxia ensues secondary to insufficient oxygen delivery due to widespread irregularities within the vascular system [19], including diminished and uneven vessel density, extended and tortuous vessels, atypical diameters, and vessel wall abnormalities [20]. These morphological abnormalities collectively augment geometric resistance to blood circulation and impair vascular function [21]. Low oxygen and its sequelae play a critical role in the pathogenesis of a broad spectrum of human diseases, especially those in which the vasculature is involved [22]. AVMs, as they are exposed to relatively hypoxic conditions due to rapid shunting, are especially believed to impact the pathophysiologic process leading to insufficient tissue oxygenation.

miRNAs, a type of ribonucleic acid (RNA) with a length of approximately 17–25 nucleotides, play a vital role in the regulation of gene expression. miRNAs do not store genetic information, however, they function as potent gene regulators by binding complementarily to various target messenger RNAs (mRNAs) to regulate gene transcription and translation. Most of miRNAs are transcribed from DNA sequences into primary miRNAs, processed into precursor miRNAs, and finally into mature miRNAs [23,24].

Recent studies reported the discovery of relevant miRNAs in patients with various diseases including cancer and have demonstrated that the degree of their expression is closely correlated with disease progression [25,26,27]. However, to date, there is limited knowledge regarding which miRNAs are involved in AVMs, including their clinical significance.

This study aimed to validate AVM-related miRNAs and report a comparative analysis between the normal and AVM endothelium under hypoxic conditions and with or without VEGF treatment.

## 2. Results

### 2.1. Analysis of Endothelial Cell miRNA Expression

#### 2.1.1. Differential Expression of Seven miRNAs in Endothelial Cells

To achieve this objective, in normal and AVM groups, we considered miRNAs strongly associated with ECs, where the differential miRNA expression exhibited a fold change of at least 1.2 between the normal and AVM groups. Statistically significant differences with a *p*-value < 0.05 were selected. All these miRNAs were differentially expressed by at least two standard deviations above the background, determined by the nCounter^®^ Human miRNA Expression Assay, in descending order. Among these miRNA candidates, five miRNAs (miR-135b-5p, miR-496, miR-132-3p, miR-193a-3p, and miR-193b-5p) displayed an increase of at least 1.2-fold, whereas two miRNAs (miR-137 and miR-30a-3p) demonstrated a decline in the AVM group compared to the normal group (Table 1).

#### 2.1.2. Upregulation of miR-135b-5p and Downregulation of miR-137 in the AVM Group

At the laboratory level, quantitative real-time PCR was also conducted to verify the expression of seven miRNA candidates in normal and AVM groups. Specifically, the relative expression levels of miR-135b-5p were remarkably increased (*p*-value = 0.0253) in the AVM group compared to the normal group. Conversely, the relative expression levels of miR-137 were significantly decreased (*p*-value = 0.0238) in the AVM group (Figure 1). Thus, we identified that among the seven miRNA candidates, miR-135b-5p and miR-137 were significant in the AVM group compared with the normal group.

### 2.2. Comparison of miRNA Expressions in Hypoxia with/without VEGF Treatment

#### 2.2.1. MicroRNA Expression under Hypoxic Conditions

Comparing groups A, B, C, and D, hypoxic conditions were associated with increased endothelial activity in AVM when compared to the normal group, leading to a marked upregulation of miR-135b-5p (*p*-value = 0.0238) (Figure 2A). 

Among the upregulated miRNAs, including miR-495, miR132-3p, miR-193a-5p, and miR-193b-5p, groups B and D revealed an increase compared to groups A and C, however, no statistical significance was noted (Figure 2D,G,J,M).

For downregulated miRNAs such as miR-137 and miR-30a-3p, group B demonstrated an increase compared to group A, and group D revealed a decrease compared to group B, although there was no statistical significance (Figure 3A,D).

#### 2.2.2. MicroRNA Expression under VEGF Treatment

Comparing groups A, B, E, and F, angiogenesis under VEGF treatment failed to demonstrate a significant increase in miR-135b-5p under normal oxygen conditions, however, a substantial increase in miR-135b-5p expression was noted in AVM following VEGF treatment (Figure 2B).

For upregulated miRNAs, including miR-495, miR132-3p, miR-193a-5p, and miR-193b-5p, groups B and F demonstrated an increase compared to groups A and E, however, no statistical significance was noted (Figure 2E,H,K,N). 

For the downregulated miRNAs, such as miR-137, group B revealed an increase compared to group A, whereas miR-30a-3p exhibited a decline, and both miRNAs showed a similar level of expression between groups E and F, neither of which reached statistical significance (Figure 3B,E).

#### 2.2.3. MicroRNA Expression under Hypoxia and VEGF Treatment

Although there was no statistical significance in the comparison between groups A and B and groups G and H, VEGF treatment appeared to downregulate miR-135b-5p in AVM under hypoxic conditions (Figure 2). Particularly, miR-193-5p was upregulated in normal ECs under hypoxic conditions, although this was statistically insignificant. Moreover, no significant difference was observed in the AVM (Figure 2D). 

For upregulated miRNAs, including miR-495, miR132-3p, and miR-193b-5p, groups B and H demonstrated a decrease when compared to groups A and G. However, an increase was observed in miR-193a-5p, although no statistical significance was observed for all cases (Figure 2J–L,O). 

In the downregulated miRNAs, such as miR-137 and miR-30a-3p, group B revealed an increase compared to group A, and group D demonstrated a decrease compared to group B, although no statistical significance was observed (Figure 3C,F).

## 3. Discussion

Among vascular malformations, AVM is the most challenging abnormality to treat and is associated with a poor prognosis. Similar to other incurable diseases, treatment and prognosis differ in AVMs; however, the current diagnosis mainly identifies the condition after progression, underscoring the need for early screening. AVM remains challenging to cure, and treatments such as sclerotherapy and surgery primarily target symptom alleviation, thus the requirement for a new treatment method. 

miRNAs, which are short, non-coding RNA molecules, play a significant role in regulating gene expression after transcription. These miRNAs are especially crucial in cardiovascular health, where they influence the formation of new blood vessels—a process known as angiogenesis. ECs, which line blood vessels, rely on these miRNAs for proper functioning. When ECs respond abnormally due to adverse changes in blood dynamics or prolonged exposure to conditions like chronic hypoxia or inflammation, it can result in improper or abnormal angiogenesis. This dysfunction can be a precursor to a variety of vascular diseases [28].

The few available models mainly predict the relationship between miRNA expression patterns and the physiological symptoms of diseases. Examples of these include the miR-193a model in ovarian cancer and models examining miRNA control circuits in the process of epithelial–mesenchymal transition. These models aim to establish a connection between the expression of miRNA-related molecules and the physical manifestations of diseases [29,30].

Recent miRNA research indicates the potential for novel diagnostic and treatment possibilities, and the extensive use of miRNA microarrays has made it possible to discover numerous microRNAs that might be used as potential biomarkers [31,32]. 

We identified that miRNAs could be associated with endothelial AVM cell functions, such as angiogenesis and hypoxia. An analysis was conducted to understand the functions of these miRNAs under hypoxic conditions and VEGF treatment, which are similar to stimuli in AVM. Seven previously identified EC-relevant miRNAs were selected [33]. The five AVM-upregulated miRNAs (miR-135b-5p, miR-496, miR-132-3p, miR-193a-5p, miR-193b-5p) and two downregulated miRNAs (miR-137 and miR-30a-3p) were differentially expressed. Among these, only miR-135b-5p exhibited statistically significant differences. Particularly, reports of miRNAs related to angiogenesis have been published. Hypoxia-inducible factor 1 α (HIF-1α), an essential transcription factor responsive to oxygen levels, comprises two subunits: HIF-1β, which is always present, and HIF-1α, whose stability is influenced by oxygen availability. Under normal oxygen conditions (normoxia), HIF-1α is quickly broken down through processes involving prolyl hydroxylases and factors inhibiting HIF-1, leading to its degradation by the von Hippel–Lindau (VHL) ubiquitin E3 ligase complex [34,35,36]. Thus, HIF-1 is suggested to play an important role in the body’s response to varying oxygen levels, with a significant impact on gene activation, particularly in processes like angiogenesis.

Liu et al. reported miR-135b-5p involvement in diabetic retinopathy [37]. Their research highlighted the association between miR-135b-5p and VHL protein, particularly noting that VHL expression was linked to angiogenesis and HIF-1α, which is a key player in adaptive responses, especially under decreased oxygen levels. Notably, miR-135b-5p was reported to inhibit VHL and promote HIF-1α expression [37,38]. Under low oxygen (hypoxia), HIF-1α is preserved, and it moves to the nucleus where it combines with HIF-1β. This complex then activates several genes, notably those responsible for producing VEGF-A and erythropoietin, which are vital for angiogenesis and red blood cell production [39,40,41,42]. Thus, miR-135b-5p, which is also actively studied in gastric and pancreatic cancers, demonstrates pathophysiological hypoxic and angiogenetic characteristics in AVM. This state may lead to relatively decreased oxygen levels in the surrounding tissue, which can be speculated to have similar mechanisms to VHL expression and HIF-1α in diabetic retinopathy. Considering the study findings, under hypoxic conditions, AVM exhibits angiogenic characteristics, therefore establishing a strong correlation with HIF-1 which plays a key role in adaptation to hypoxia. The oxygen-regulated alpha subunit (HIF-1α) is not associated with VEGF and mechanisms in cancer-related studies. Although many of the molecular components that are involved in the miR control of the HIF-VEGF pathway in ECs have been characterized, the detailed dynamics of how they mechanistically interact with each other within the signaling network are poorly understood [43]. In summary, the intricate relationship between miRNAs, hypoxic conditions, and the VEGF pathway plays a crucial role in the pathophysiology of AVMs and other vascular diseases. The regulation of angiogenesis by miRNAs under hypoxic conditions, especially through the HIF-VEGF axis, opens potential avenues for therapeutic interventions targeting these molecular pathways.

This study showed that miR-135b-5p, an important microRNA, regulates gene expression involved in protein synthesis via mRNA and was significantly increased under hypoxic stimulus, within the AVM, which possesses angiogenetic and hypoxic characteristics, compared to normal vessels under the study conditions. The statistically significant increase in expression was noted not only when subjected to hypoxic conditions but also under VEGF treatment, which indicates its connection to VEGF and HIF-1α. However, when hypoxic conditions and VEGF treatment were administered, the decline in the expression might lead to potential adverse effects due to VEGF overexpression, potentially induced by HIF-1α overexpression. Further research is warranted to explore the possibility of adverse effects and to better understand the interaction between HIF-1a, VEGF, and miR-135b-5p in the context of hypoxia or/and VEGF treatment. VHL is considered a target recruitment subunit of an E3 ubiquitin ligase complex that recruits hydroxylated HIF-α for proteasomal degradation under normoxia, and VEGF contributes to retinal changes under hypoxic environments [44,45,46].

Analyzing the results, hypoxic conditions resulted in an increased endothelium in AVM compared to normal conditions, leading to an increase in miR-135b-5p (Figure 2A). Under normal conditions, angiogenesis under VEGF treatment failed to show a significant increase in miR-135b-5p; however, a substantial increase was noted in the AVM (Figure 2B). We have observed an increase in miR-135b-5p within the AVM, suggesting its pivotal role in the pathophysiological process of AVM. Thus, the results indicate the elevation of miR-135b-5p in AVM might contribute to disease exacerbation by interacting with hypoxia. The VEGF application also demonstrated an increase in AVM, indicating disease progression.

Although there was no statistical significance, miR-135b-5p levels decreased under the combined influence of hypoxia and VEGF compared to hypoxia alone in group D. This result raises the possibility that hypoxia may have a more pronounced impact on disease exacerbation than VEGF, or alternatively, VEGF could influence the hypoxic environment by impeding it or producing an adverse effect (Figure 2C). Thus, it is likely that future studies need to explore the interaction between hypoxia and VEGF.

Furthermore, miR-193a-5p was upregulated in normal ECs under hypoxic conditions. However, no significant difference was observed in AVM. Chang et al. reported a significant upregulation of miR-193-3p in vascular tissues in an ischemia–reperfusion injury rat model experiment, and Fang et al. discovered the upregulation of miR-193-5p in the circulatory systems of human patients with hypertrophic cardiomyopathy, indicating it might be a potential biomarker for diffuse myocardial fibrosis [47,48]. An animal experiment by Yi et al. elucidated that miR-193a-5p modulated angiogenesis via insulin-like growth factor 2 in type 2 diabetic cardiomyopathy [49]. However, several previous studies have shown the aberrant expression of the miR-193 family in diabetic or cardiac diseases, and the functional mechanisms of miR-193 have not been fully explored. Yi et al. suggested miR-193-5p may functionally regulate angiogenesis in diabetic cardiomyopathy. This indicates that miR-193b-5p might have an impact on AVM pathophysiology. Thus, miR-193-5p is considered a necessary candidate for further research (Figure 2). Neither downregulated microRNA-137 nor 30a-3p revealed statistically significant results or consistent trends (Figure 3). 

There were several limitations in our study. First, this study suggests that miR-135b-5p is highly associated with the pathogenesis of AVM, however, clear evidence for this conclusion needs to be clarified through further study including miR-135b-5p loss or gain-of-function analysis. Second, the highly variable data may be attributed to the limited sample size, although the variance was minimized through planned operation. However, this study still holds significance from the fact it compiled and analyzed rare cases. Further study will continue with additional clinical data and long-term follow-up. 

Based on these results, it can be concluded that miR-135b-5p plays an important role in the pathophysiologic process in AVM and may have applications as a biomarker to establish diagnosis and prognosis and potentially extend to therapeutic interventions.

## 4. Materials and Methods

### 4.1. MicroRNA Profiling and Validation in Endothelial Cells

This was a prospective single-center study. The institutional review board at Kyungpook National University Hospital approved the study protocol (Approval No.: 2023-04-004-002). The study was performed in accordance with the Declaration of Helsinki guidelines. A total of 23 patients (10 patients without AVM and 13 patients with AVM) were enrolled in the study. 

#### 4.1.1. Isolation and Culture of Endothelial Cells

After patients’ consent, AVM tissues (*n* = 13) were collected during surgical resection of the AVM lesions, and normal vasculature tissues (*n* = 10) were collected from discarded normal subcutaneous tissues including normal arterial vasculature during other surgeries (Table 1). The surgical samples were rinsed with phosphate-buffered saline (pH 7.4, LB004-02, Welgene, Gyeongsan, Republic of Korea). The tissues were cut into small pieces using a surgical scalpel and scissors. To separate the epidermis and dermis, the tissues were incubated in Dispase II (Gibco^TM^ Dispase 17105-041, Thermo Fisher Scientific Korea, Seoul, Republic of Korea) for 24 h at 4 °C. After incubation, the dermal layer was collected and settled in 10 mL of Hanks’ balanced salt solution (14170-112, Gibco). The supernatant was removed and 5 mL of collagenase type I (4196, Worthington, OH, USA) was added. The sample was incubated at 37 °C, 170 bpm in a shaking incubator for 1 h. Approximately 10 mL EMB-2 media (cc-3156, Lonza, Basel, Switzerland) was added and was passed through a 70 μm nylon filter. After centrifugation (1000 rpm, 5 min), cells were grown in EMB-2 media and were maintained in an incubator at 37 °C with room air oxygen levels and 5% CO_2_.

#### 4.1.2. MicroRNA Profiling

miRNAs were extracted from ECs isolated from human-derived cells (normal group and AVM group) using QIAzol lysis reagent (Qiagen, Valencia, CA, USA) as described in our previous study [25]. Quantity and quality controls of miRNAs were assessed using a Denovix 11 AATI Fragment Analyzer (Denovix Inc., Wilmington, DE, USA) to ascertain data integrity.

Approximately 800 human miRNAs from 23 human-derived ECs (normal = 10 and AVM = 13) were investigated using the nCounter^®^ Human miRNA Expression Code Set, version 3b (NanoString Technologies, Inc., Seattle, WA, USA) according to the manufacturer’s protocol. Data normalization for 23 samples was performed with positive control and housekeeping genes included in the code set using nSolver™ 4.0 software analysis (NanoString Technologies, Inc.). The miRNA profile analysis results were considered statistically significant with fold change ± 1.2, with a *p*-value < 0.05 (Table 2).

#### 4.1.3. MicroRNA Validation

Differential expressions of five upregulated and two downregulated miRNAs in the miRNA profile analysis were validated by performing quantitative real-time polymerase chain reaction (PCR) using TaqMan microRNA assays (Applied Biosystems, Foster City, CA, USA). All values were measured in triplicate. The relative expressions of seven miRNAs were normalized through *RNU6B* and calculated using the 2^−ΔΔCt^ method.

### 4.2. Comparison of miRNAs’ Expression in Hypoxia with/without VEGF Treatment

A total of 14 samples (6 samples without AVM and 8 samples with AVM) were enrolled in the study. 

Our experiment was divided into eight groups for comparison of miRNAs as below. Group A: endothelial cells from Nor + nor O_2_ + w/o VEGF (*n* = 6);Group B: endothelial cells from AVM + nor O_2_ + w/o VEGF (*n* = 8);Group C: endothelial cells from Nor + hypo O_2_ + w/o VEGF (*n* = 6);Group D: endothelial cells from AVM + hypo O_2_ + w/o VEGF (*n* = 8);Group E: endothelial cells from Nor + nor O_2_ + w VEGF (*n* = 6);Group F: endothelial cells from AVM + nor O_2_ + w VEGF (*n* = 8);Group G: endothelial cells from Nor + hypo O_2_ + w VEGF (*n* = 6); Group H: endothelial cells from AVM + hypo O_2_ + w VEGF (*n* = 8).

Nor: human normal vessels, AVM: human AVM vessels, nor O_2_: normoxia, hypo O_2_: hypoxia, w/o VEGF: without VEGF, w VEGF: with VEGF. 

#### 4.2.1. Hypoxic Conditions

The hypoxic conditions used in the experiments consisted of 1% O_2_ and 5% CO_2_ at 37 °C. A hypoxia chamber (Modular Incubator Chamber, Billups-Rothenberg, Inc., San Diego, CA, USA) was used to maintain the hypoxic partial pressure. The cultured ECs from AVM tissues and normal tissues were exposed to hypoxic conditions for 14 h. 

#### 4.2.2. VEGF Treatment

ECs (1 × 10^6^) from AVM tissues and normal tissues were incubated in serum-free medium with 10 ng/mL of VEGF (sc-152, Santa Cruz Biotechnology Inc., Santa cruz, CA, USA) for 14 h at 37 °C under normoxic and hypoxic conditions.

#### 4.2.3. MicroRNA Expression

We conducted a miRNA assay following the method described in Section 4.1.2 and Section 4.1.3 to analyze the relative expressions of miRNAs.

#### 4.2.4. Statistical Analysis

The data were presented as the mean ± standard deviation of three experimental replicates. To evaluate differences in miRNA expression levels between groups, paired Student’s *t*-test and analysis of varianceone-way ANOVA were employed. Statistical significance was set to a *p*-value < 0.05. All statistical analyses were conducted using SPSS version 22.0 (Chicago, IL, USA).

## Figures and Tables

**Figure 1 ijms-25-04888-f001:**
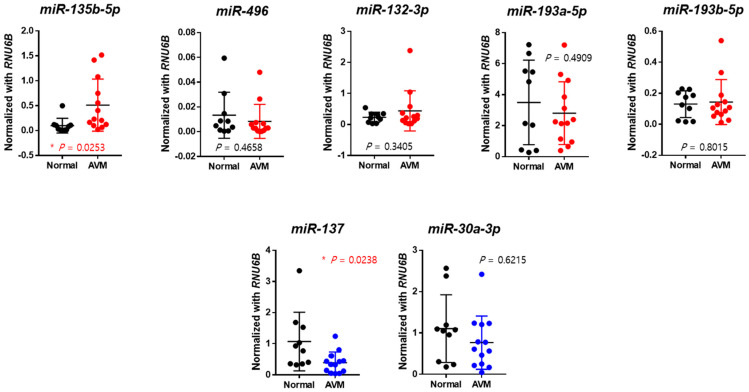
Validations of the normal group and AVM group using real-time polymerase chain reaction. Each *y*-axis within the figure is normalized to *RNU6B.* Micro RNA-137 showed a statistically significant lower expression in the endothelial cells of AVM vessels compared to those in normal vessels (*p*-value = 0.0238). * *p*-value < 0.05.

**Figure 2 ijms-25-04888-f002:**
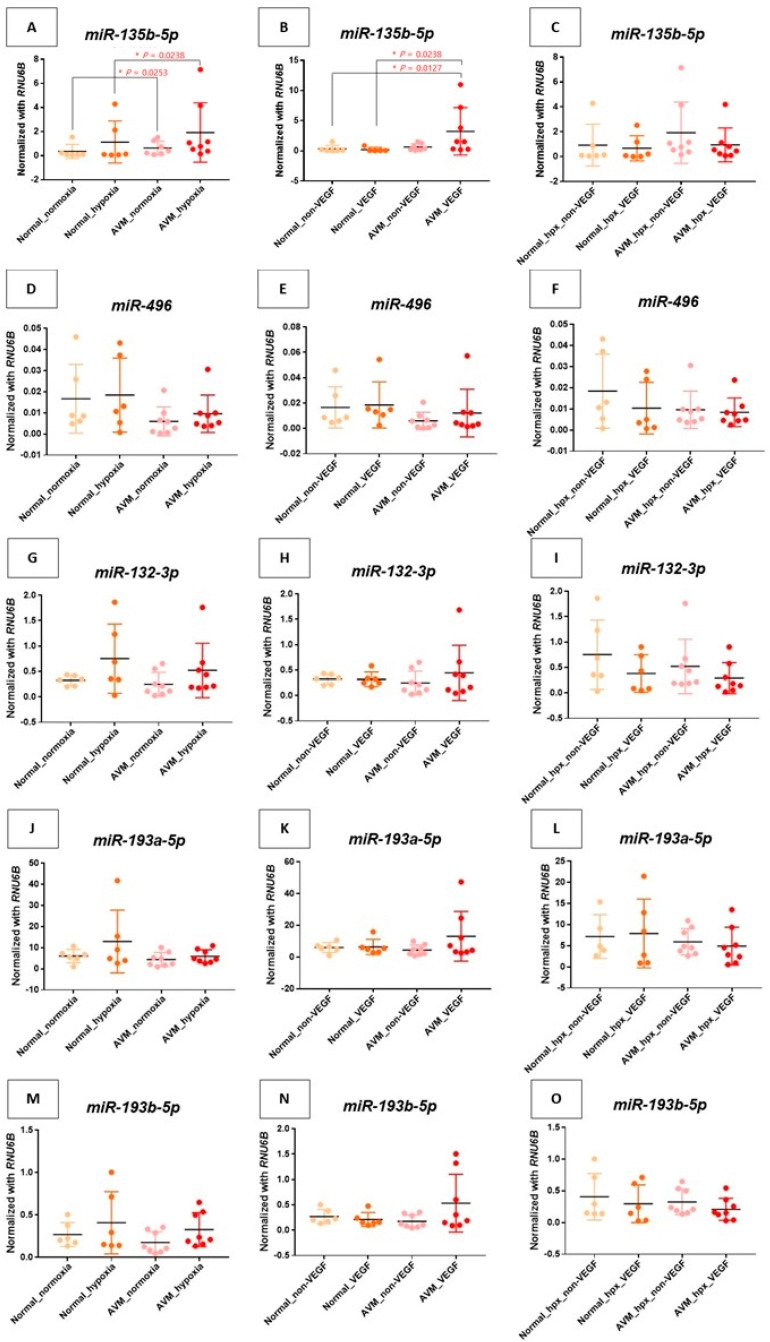
Analysis of upregulated miRNAs in endothelial cells in various environments. Each *y*-axis within the figure is normalized to *RNU6B.* * *p*-value < 0.05.

**Figure 3 ijms-25-04888-f003:**
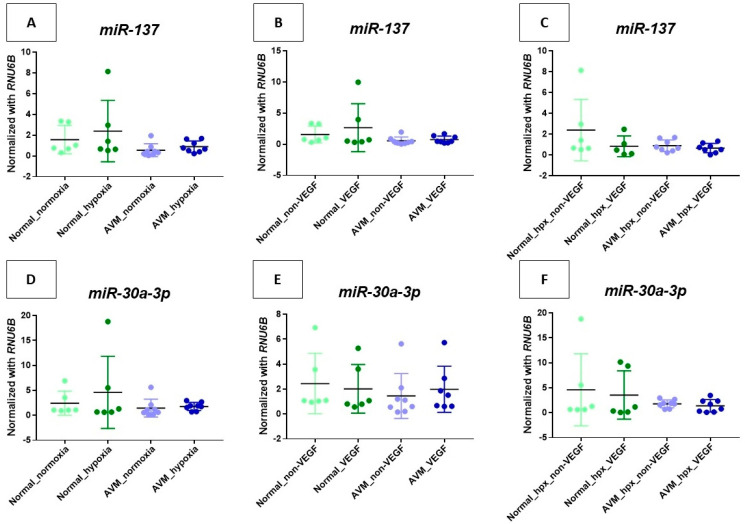
Analysis of downregulated miRNAs in endothelial cells in various environments. Each *y*-axis within the figure is normalized with *RNU6B*.

**Table 1 ijms-25-04888-t001:** Patient characteristics.

No.	Diagnosis	Sex	Age	Underlying Disease	Oxygen Condition	VEGF Condition
1	Normal	Male	45	HTN		
2	AVM	Male	18	-	Normoxia/Hypoxia	Treated/ Non-treated
3	Normal	Female	43	-	Normoxia/Hypoxia	Treated/ Non-treated
4	Normal	Male	60	-	Normoxia/Hypoxia	Treated/ Non-treated
5	AVM	Male	33	-	Normoxia/Hypoxia	Treated/ Non-treated
6	AVM	Female	1	-		
7	Normal	Female	43	-	Normoxia/Hypoxia	Treated/ Non-treated
8	AVM	Female	13	-		
9	AVM	Male	22	-		
10	AVM	Female	26	-	Normoxia/Hypoxia	Treated/ Non-treated
11	AVM	Male	4	-		
12	Normal	Male	31	-	Normoxia/Hypoxia	Treated/ Non-treated
13	Normal	Male	20	-		
14	AVM	Female	16	-	Normoxia/Hypoxia	Treated/ Non-treated
15	AVM	Male	25	-	Normoxia/Hypoxia	Treated/ Non-treated
16	AVM	Male	26	-		
17	Normal	Female	62	HTN		
18	AVM	Male	38	-	Normoxia/Hypoxia	Treated/ Non-treated
19	Normal	Male	45	-	Normoxia/Hypoxia	Treated/ Non-treated
20	AVM	Male	24	-	Normoxia/Hypoxia	Treated/ Non-treated
21	Normal	Male	21	-	Normoxia/Hypoxia	Treated/ Non-treated
22	AVM	Male	46	HTN	Normoxia/Hypoxia	Treated/ Non-treated
23	Normal	Female	47			

**Table 2 ijms-25-04888-t002:** The differential expression of miRNAs in endothelial cell profiling. miRNAs from 23 human-derived endothelial cells were assessed as miRNA candidates through profiling data. miRNA profiles were analyzed and considered statistically significant with a fold change of ±1.2 and a *p*-value < 0.05.

Upregulated miRNA	Fold Change (AVM/Normal)	*p*-Value *
miR-135b-5p	4.81	0.0484
miR-496	2.02	0.0237
miR-603	1.80	0.0213
miR-132-3p	1.73	0.0274
miR-193a-5p + has-miR-193b-5p	1.42	0.0315
Downregulated miRNA	Fold Change (AVM/normal)	*p*-value *
miR-30a-3p	−1.45	0.0212
miR-137	−2.54	0.0185

* AVM/normal, paired *t*-test.

## Data Availability

The data presented in this study are available in the manuscript.

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
