# Peer review of "MicroRNA-135b-5p Is a Pathologic Biomarker in the Endothelial Cells of Arteriovenous Malformations"

_ijms, 2024, doi:10.3390/ijms25094888_

Round 1

Reviewer 1 Report (New Reviewer)

Comments and Suggestions for Authors

Major Comments:

Abstract:

1. In the final sentence of the abstract (Line 22-26), authors reported that miR-135b-5p is implicated in the pathophysiological processes of AVM but the specific mechanisms involved are not mentioned. Authors should briefly mention the potential mechanisms through which miR-135b-5p influences AVM pathophysiology.

2. The statistical significance is mentioned (p-value=0.0238) in abstract. However, It would be helpful if authors provide the statistical tests used.

Introduction:

Authors stated in introduction, "All types of vascular anomalies manifest similar surface characteristics...” This statement is may not be accurate and authors should re-evaluate about this statement and improved the information according to clinical presentation and characteristics of different vascular anomalies.

Authors explained the hypoxia and its role in AVM pathogenesis as hypoxia is a critical factor in AVM development, but it can be improved by including its effects on cellular signaling pathways and angiogenesis.

Materials and Methods:

1. The sample size justification is unclear. Authors are suggested to provide the basis of sample size calculations.  It would be helpful if authors provide a rationale for selecting 23 patients (10 without AVM and 13 with AVM).

2. The methods for miRNA profiling and validation appear sound, but additional information on the selection criteria for miRNAs of interest and the rationale behind their selection would enhance the clarity and transparency of the study.

3. Authors used analysis of variance (ANOVA) tool for statistical analyses but does not specify the specific ANOVA model employed.

Discussion:

1. The discussion on the role of miR-135b-5p in AVM pathophysiology is insightful but it lacks specificity regarding the underlying mechanisms. Authors should elaborating the potential molecular pathways through which miR-135b-5p influences AVM progression.

2. The discussion of miR-193a-5p and its relevance to AVMs is intriguing but lacks integration with the overall findings of the study. Authors should integrate their findings to the miRNA dysregulation in AVMs and compare and contrast with recent published studies.

3. Authors are suggested to discuss the limitations as well.

General Comments:

1. Proof read the article for grammatical and typographical errors.

2. Provide abbreviations list used in manuscript.

Comments on the Quality of English Language

Proof read the article for grammatical and typographical errors.

Author Response

<Reviewer 1>

1. Summary

2. Point-by-point response to Comments and Suggestions for Authors

Abstract:

  1. In the final sentence of the abstract (Line 22-26), authors reported that miR-135b-5p is implicated in the pathophysiological processes of AVM but the specific mechanisms involved are not mentioned. Authors should briefly mention the potential mechanisms through which miR-135b-5p influences AVM pathophysiology.

Response : Based on the summarized results, the sentence was written to intend that miR-135b-5p may be involved in the pathophysiological process of AVM. However, it seems there was an error in using a rather assertive expression. We have revised the expression on line 29-31.

  1. The statistical significance is mentioned (p-value=0.0238) in abstract. However, It would be helpful if authors provide the statistical tests used.

Response : We agree your kind comment and have added the test we have went through on line 22 and 368

Introduction:

Authors stated in introduction, "All types of vascular anomalies manifest similar surface characteristics...” This statement is may not be accurate and authors should re-evaluate about this statement and improved the information according to clinical presentation and characteristics of different vascular anomalies.

Response : The indicated statement had an error, we have corrected on line 35

Authors explained the hypoxia and its role in AVM pathogenesis as hypoxia is a critical factor in AVM development, but it can be improved by including its effects on cellular signaling pathways and angiogenesis.

Response : There are some references that hypoxia and local ischemia was a common condition reported in cerebral AVM and decrease in tissue oxygen concentrations has been considered as the leading cause of endothelial cell proliferation  however, to the best of our knowledge there are no reported cases suggesting the improvement by its cellular signaling pathway. We will incorporate it into this paper and future research.

Materials and Methods:

  1. The sample size justification is unclear. Authors are suggested to provide the basis of sample size calculations.  It would be helpful if authors provide a rationale for selecting 23 patients (10 without AVM and 13 with AVM).

Response : Our team always believes that in experimental clinical studies, it is essential to convey facts and share the results with others conducting similar research. The number of samples from each tissue that could be collected through surgery during the same period was accurately reflected. We were preparing to conduct the experiment with consistent numbers, but due to issues with the sample storage device, the number of samples slightly decreased (10 to 6 and 13 to 8, each) This reduction has been accurately noted. Any potential confusion was clarified by presenting this information in the patient characteristics table. We are very grateful for your comment on this aspect.

  1. The methods for miRNA profiling and validation appear sound, but additional information on the selection criteria for miRNAs of interest and the rationale behind their selection would enhance the clarity and transparency of the study.

Response : We apologize for not making it clear. We set the reference to fold change 1.2 by the following articles but did not separately indicate it in the text.

In the study by He Q et al. titled "Exosomal miR-3131 derived from endothelial cells with KRAS mutation promotes EndMT by targeting PICK1 in brain arteriovenous malformations" (CNS Neurosci Ther. 2023;29(5):1312-1324), It set a significant fold change of 1.2 among different KRAS groups to differentiate the responsible exosomal microRNA.

Similarly, in the research by Schaefer A et al. titled "Suitable reference genes for relative quantification of miRNA expression in prostate cancer" (Exp Mol Med. 2010;42(11):749-758), They also used a fold change of 1.2 as a criterion to assess significant miRNA candidates for prostate cancer.

Thank you for your valuable feedback.

  1. Authors used analysis of variance (ANOVA) tool for statistical analyses but does not specify the specific ANOVA model employed.

Response : We apologize for not describing clearly the statistical methods. Following the reviewer's comment, we have edited this to state that we used the one-way analysis of variance (ANOVA) test.

Discussion:

  1. The discussion on the role of miR-135b-5p in AVM pathophysiology is insightful but it lacks specificity regarding the underlying mechanisms. Authors should elaborating the potential molecular pathways through which miR-135b-5p influences AVM progression.

Response : Thank you for the sharp review. Our current study was initiated upon observing significant changes in miR-135b-5p among the seven miRNAs selected under hypoxia and VEGF conditions. The issues you mentioned are indeed considered limitations in our research, and we plan to address them in future studies to achieve clearer results. Thank you.

  1. The discussion of miR-193a-5p and its relevance to AVMs is intriguing but lacks integration with the overall findings of the study. Authors should integrate their findings to the miRNA dysregulation in AVMs and compare and contrast with recent published studies.

Response : In the blood samples at the cellular level, miR-193a-5p was found upregulated. Although there have been no relevant previous reports to this until now, it showed significant results in our experiments, suggesting its potential significance. We are preparing to include it in our future studies for further investigation. While there haven't been definitive findings yet, we are planning to summarize our results in the future.

  1. Authors are suggested to discuss the limitations as well.

Response : Thank you for kindly reviewing our paper despite its shortcomings. We have noted the limitations on page 8, lines 281-290, and will consider adding any additional limitations that need to be addressed

General Comments:

  1. Proof read the article for grammatical and typographical errors.

Response: The manuscript has been revised through a professional English proofreading service. Thank you.

  1. Provide abbreviations list used in manuscript.

Response: We have prepared a List of Abbreviations as follows. Please let me know where to put it in the Manuscript and I will fill it out. Thank you

List of Abbreviations

AVM: Arteriovenous Malformations

ECs: Endothelial Cells

miRNA: MicroRNA

VEGF: Vascular Endothelial Growth Factor

PCR: Polymerase Chain Reaction

VHL: Von-Hippel-Lindau

HIF-1α: Hypoxia-Inducible Factor 1 Alpha

HIF-1β: Hypoxia-Inducible Factor 1 Beta

VEGF: Vascular Endothelial Growth Factor

EPO: Erythropoietin

PHD: Prolyl Hydroxylase

FIH: Factors Inhibiting HIF-1

EMB: Endothelial Cell Growth Medium

Reviewer 2 Report (New Reviewer)

Comments and Suggestions for Authors

Reviewer comments & Suggestions

MicroRNAs (miRNAs) are a group of small non-coding RNAs that are involved in regulating a range of developmental and physiological processes; their dysregulation has been associated with the development of diseases including cancer, therefore Some miRNAs have been validated as diagnostic or prognostic biomarkers in various diseases. AVM is a serious brain disorder and possibly treated with radiation therapy or by surgery. In this paper authors have observed the miRNA that are upregulated or downregulated in AVM tissue of patient as compare to normal. Final result concluded the miR-135b-5p was significantly upregulated in the AVM compared to that in normal conditions, corresponding to increased endothelial activity.

Research articles are scientifically sound, but have many shortcomings. A major revision is needed.

Scientific comments

1.      VEGF is a critical regulator of vascular function and has a potential impact on AVM pathogenesis, provide some more supportive data (citation) for this.

2.      Mentioned at least a few major targeted RNA of miR-135b-5p related to AVM pathogenesis.

3.      Lines 246-248, is not well understanding. either re frame the sentence or explain little more.

4.      Are all patients have gone through surgery? The one and four-year-old patient too?

5.      For the study, do have you selected a specific age group? There are 2 patients of age of 1 and 4? In general AVM occur at middle/old age.

6.      Sample size is very small. Have you calculated the power of sample?

7.      How you calculated the concentration of VEGF for treatment? Growth factors are present in pg concentration in body fluid.  Is it already mentioned in previous studies? If yes, provide citation.

8.      Lines 169-172, are not providing any core information. Discuss how this studied miRNA regulate the specific mRNA function and plays crucial role in AVM pathogenesis.

Minor & typo errors

1.      Line 53, give a space b/w ECs and require, IN line 284 & 286, give a space b/w full stop and the.

2.      Line 85-87, you have mentioned recent studies and provide only one citation. At least provide 4-5 citations for this statement.

3.      Line 189, 214, 269 delete extra full stop.

4.      Line 245, delete full stop after AVM

5.      Line 240, check the citation? Is it 38-40?

6.      Line 274, check the spelling of including.

7.      Line 281, add full stop after completion of line.

8.      Line 347, add full stop after partial pressure.

9.      Line 355, is this sentence incomplete? If yes. Replace comma with full stop

10.  Check the reference format and rewrite accordingly.

Comments on the Quality of English Language

A minor English editing is required. 

Author Response

<Reviewer 2>

1. Summary

2. Point-by-point response to Comments and Suggestions for Authors

Scientific comments

  1. VEGF is a critical regulator of vascular function and has a potential impact on AVM pathogenesis, provide some more supportive data (citation) for this.

Response : I agree with your comment. I have found and added an appropriate citation for the content on page 2, lines 65-71. Thank you.

  1. Mentioned at least a few major targeted RNA of miR-135b-5p related to AVM pathogenesis.

Response : messenger RNA(mRNA) is a type of RNA used in the process of transcription from DNA to RNA, and it is involved in synthesizing proteins within the cell, and microRNAs are short RNA molecules that primarily regulate and control messenger RNA. MicroRNAs interact with messenger RNA to either inhibit or promote protein synthesis.

Research into the roles between messenger RNA and microRNA is actively ongoing, but the specific target mRNA of certain miRNAs remains somewhat unknown. Our study also discovered that indirectly, through the upregulation or downregulation of miRNAs, they significantly influence pathways. However, we have yet to identify the target mRNA of miR-135b-5p or find references for it. We will include this in further research. Thank you

  1. Lines 246-248, is not well understanding. either re frame the sentence or explain little more.

Response :  Thank you, we agree your opinion and corrected the obscure sentence on page 7 line 254-258.

  1. Are all patients have gone through surgery? The one and four-year-old patient too?

Response : Yes, they have all went through surgery under general anesthesia.

  1. For the study, do have you selected a specific age group? There are 2 patients of age of 1 and 4? In general AVM occur at middle/old age.

Response : To prevent bias, we did not set a specific age group but conducted our study on patients who underwent surgery within the same period. Although AVM is prevalent in middle-aged and elderly individuals, the two cases involved opted for early operation after discussing the prognosis and potential complications with their guardians.

  1. Sample size is very small. Have you calculated the power of sample?

Response : Thank you for your valuable comments. The majority of the samples were collected and processed by the planned schedule, usually within the same timeframe. The rarity of the disease posed challenges in accumulating a large number of samples and we did not calculate the power of sample. In future studies, we aim to address these considerations more precisely as you mentioned. We will continue our research efforts to accumulate the sample size.

  1. How you calculated the concentration of VEGF for treatment? Growth factors are present in pg concentration in body fluid.  Is it already mentioned in previous studies? If yes, provide citation.

Response : We experimented with a VEGF treatment concentration of 150 pg/ml, which corresponds to 50% of the maximum effect, as reported in the paper by Beheshtizadeh et al. We apologize for any confusion caused by not providing a clear reference citation.

  1. Lines 169-172, are not providing any core information. Discuss how this studied miRNA regulate the specific mRNA function and plays crucial role in AVM pathogenesis.

Response : As mentioned above, although we couldn't find a specific mRNA or its exact function, we will actively incorporate this into our next research. Thank you for the valuable suggestion.

Minor & typo errors

  1. Line 53, give a space b/w ECs and require, IN line 284 & 286, give a space b/w full stop and the.

Response : Thank you for your delicate review. We gave the space on indicated line (currently line 56, 282 &286)

  1. Line 85-87, you have mentioned recent studies and provide only one citation. At least provide 4-5 citations for this statement.

Response : Thank you for the valuable feedback. As you suggested, We have supplemented the content with supporting citations on line 90

  1. Line 189, 214, 269 delete extra full stop.
  2. Line 245, delete full stop after AVM

Response : Thank you for reviewing thoroughly. I have made the revisions you mentioned.

  1. Line 240, check the citation? Is it 38-40?

Response : Yes, we have corrected it on line 248

  1. Line 274, check the spelling of including.

Response : Thank you (line 283)

  1. Line 281, add full stop after completion of line.
  2. Line 347, add full stop after partial pressure.
  3. Line 355, is this sentence incomplete? If yes. Replace comma with full stop

 Response : We have added the missing full stop, thank you.

  1. Check the reference format and rewrite accordingly.

 Response : Thank you.

Round 2

Reviewer 1 Report (New Reviewer)

Comments and Suggestions for Authors

Authors have diligently addressed almost all the concerns raised during the review process. The revisions made have significantly improved the quality of the article. However, authors may add abbreviation list provided in the comments' response at the end of the manuscript. 

Author Response

Reviewer 1.

Comments and Suggestions for Authors

Authors have diligently addressed almost all the concerns raised during the review process. The revisions made have significantly improved the quality of the article. However, authors may add abbreviation list provided in the comments' response at the end of the manuscript.

Response : Thanks to the thorough review, the quality of the paper has greatly improved and become clearer. I have added the following list of Abbreviations after the Conflicts of Interest at the end of the manuscript. This has also made it less confusing for readers regarding the abbreviations. Thank you.

List of Abbreviations: AVM, Arteriovenous Malformations; ECs, Endothelial Cells; miRNA, MicroRNA; VEGF, Vascular Endothelial Growth Factor; PCR, Polymerase Chain Reaction; VHL, Von-Hippel-Lindau; HIF-1α, Hypoxia-Inducible Factor 1 Alpha; HIF-1β, Hypoxia-Inducible Factor 1 Beta; VEGF, Vascular Endothelial Growth Factor; EPO, Erythropoietin; PHD, Prolyl Hydroxylase; FIH, Factors Inhibiting HIF-1; EMB, Endothelial Cell Growth Medium

Reviewer 2 Report (New Reviewer)

Comments and Suggestions for Authors

Reviewer comments & Suggestions

1.      The authors have justified all the comments very well.

2.      The abstract is updated as suggested.

Minor comments

1.      Most typo errors are corrected however few are still in it.

2.      Line 235, ‘is a gene regulator in the generation of creating proteins via mRNA’, rewrite this sentence.

3.      Line 259-261, ‘However, under the combined influence of hypoxia and VEGF, although ……….. in group D’, is very confusing. Rewrite the statement.

4.      References pattern is still not the same. Check the reference format and rewrite accordingly.

Author Response

Reviewer 2.

Comments and Suggestions for Authors

Reviewer comments & Suggestions

  1. The authors have justified all the comments very well.
  2. The abstract is updated as suggested.

Response : Thank you very much for your positive review.

Minor comments

  1. Most typo errors are corrected however few are still in it.

Response : We have further revised the spacing, punctuation, and typographical errors. Thank you.

  1. Line 235, ‘is a gene regulator in the generation of creating proteins via mRNA’, rewrite this sentence.

Response : Thank you for the feedback. 'Regulates gene expression' accurately describes the function of miR-135b-5p, and 'involved in protein synthesis' clearly focuses on the process of protein synthesis via mRNA, allowing me to rewrite the sentence more clearly.

  1. Line 259-261, ‘However, under the combined influence of hypoxia and VEGF, although ……….. in group D’, is very confusing. Rewrite the statement.

Response : Thank you. By starting with 'although' to clarify the lack of statistical significance first, and then describing the main results, we have revised the sentence to be concise and easier to understand.

  1. References pattern is still not the same. Check the reference format and rewrite accordingly.

Response : Thank you, we have referred to the reference guideline and rewrote accordingly.

This manuscript is a resubmission of an earlier submission. The following is a list of the peer review reports and author responses from that submission.

Round 1

Reviewer 1 Report

Comments and Suggestions for Authors

Regarding the scientific content and rigor of the text, there are several points that could be addressed:

-The text should ensure that the terms and concepts used align with the current scientific consensus. For example, the naming convention of microRNAs like "miR-30a-3p" needs to be accurate throughout.

-The mechanisms described, such as those relating to VEGF and its role in vascular malformations, should be clear and precise: each statement about physiological processes should be supported by references to experimental data or established scientific theory.

-It's crucial to place findings within the context of existing literature. If the text discusses novel findings about miRNAs in relation to AVM, it should also discuss how these findings relate to or expand upon existing knowledge.

-In the discussion part around miRNAs and their potential therapeutic implications for AVM, consideration of the challenges and opportunities in translating these findings, from the bench to the bedside, should be included.

I'll be pleased to revise the next version of the manuscript.

Comments on the Quality of English Language

The sentence structure occasionally becomes convoluted, making it difficult to follow. For instance, "Tissue oxygenation is governed by a balance between 02 supply, delivered by the vasculature, and demand by metabolic outputs of tissues." could be rephrased for clarity: "Tissue oxygenation depends on a balance between oxygen supply, delivered by the vasculature, and the metabolic demand of tissues." "mik-30a-3p" should be "miR-30a-3p" if referring to the microRNA nomenclature. Phrases such as "which is similar stimuli to AVM" could be refined to "which are similar to stimuli in AVM" for clarity.

There is some redundancy in explaining the role of miRNAs, VEGF, and hypoxia that could be streamlined for brevity and impact.

The text fluctuates between "miR" and "mik" (e.g., "mik-30a-3p" should likely be "miR-30a-3p" to maintain consistency with standard miRNA nomenclature).

Use of "02" instead of "O2" for oxygen is inconsistent with scientific norms.

The text contains punctuation errors, such as missing commas which could lead to ambiguous meanings.

Author Response

-The text should ensure that the terms and concepts used align with the current scientific consensus. For example, the naming convention of microRNAs like "miR-30a-3p" needs to be accurate throughout.

>Thanks to you, I was able to reconfirm the consistency of terms and concepts throughout the entire text of the paper. I appreciate it.

-The mechanisms described, such as those relating to VEGF and its role in vascular malformations, should be clear and precise: each statement about physiological processes should be supported by references to experimental data or established scientific theory.

> Thank you for the sharp observation. There was no reference clearly specifying the relationship between VEGF and the broader spectrum of vascular malformation diseases. If this paper is fortunate enough to be published, it is anticipated to serve as a contributing reference for explaining that relationship. Instead, references discussing the relationship between VEGF and orbital cavernous malformation as well as lymphatic malformation are included in the discussion. Additional information on VEGF, which seemed lacking in general explanation, has been added on line 73-77. “Vascular endothelial growth factor (VEGF) is an effective inducer of angiogenesis and was initially characterized as a crucial growth factor for vascular endothelial cells. It’s up-regulation is observed in numerous tumors, and its role in facilitating tumor angiogenesis is clearly established [14].” Thank you.

-It's crucial to place findings within the context of existing literature. If the text discusses novel findings about miRNAs in relation to AVM, it should also discuss how these findings relate to or expand upon existing knowledge.

>Thank you for the valuable advice. To analyze the relationship between miRNA and endothelial cell function in AVM, we conducted experiments with miRNA under VEGF and hypoxic conditions. Among them, miR-135b-5p showed statistical significance. Importantly, this finding aligns with the reference mentioned in the discussion, which associates miR-135b-5p with angiogenesis through its significant relevance to the Von Hippel-Lindau protein (VHL). In our experiments, the hypoxic condition led to an increase in endothelial cell proliferation in AVM, accompanied by an elevation in miR-135b-5p levels. Similarly, under the VEGF treatment condition, AVM exhibited a significant increase in miR-135b-5p.

The sentence structure occasionally becomes convoluted, making it difficult to follow. For instance, "Tissue oxygenation is governed by a balance between 02 supply, delivered by the vasculature, and demand by metabolic outputs of tissues." could be rephrased for clarity: "Tissue oxygenation depends on a balance between oxygen supply, delivered by the vasculature, and the metabolic demand of tissues." "mik-30a-3p" should be "miR-30a-3p" if referring to the microRNA nomenclature. Phrases such as "which is similar stimuli to AVM" could be refined to "which are similar to stimuli in AVM" for clarity.

>We deeply agree with your sharp point outs. We have made corrections throughout sentences, especially those convoluted or hindered by unnecessary passive constructions, including sentences you provided as an example, to enhance overall clarity

There is some redundancy in explaining the role of miRNAs, VEGF, and hypoxia that could be streamlined for brevity and impact.

> We went through overall corrections to make sentence more brief and clear, thank you.

The text fluctuates between "miR" and "mik" (e.g., "mik-30a-3p" should likely be "miR-30a-3p" to maintain consistency with standard miRNA nomenclature).

Use of "02" instead of "O2" for oxygen is inconsistent with scientific norms.

The text contains punctuation errors, such as missing commas which could lead to ambiguous meanings.

> We have completed the revision of entire sentences to maintain consistency and follow standar nomenclature, and corrected punctuation errors. We apologize for inconvenience caused by errors and incompleteness in your review 

Reviewer 2 Report

Comments and Suggestions for Authors

            Lee et al have examined expression of microRNAs in endothelial samples from normal and AVF patients. They examined untreated samples and samples treated with hypoxia, VEGF, and hypoxia + VEGF. While the idea is of interest, unfortunately, there are serious problems with experimental design, analysis and presentation.

1.Most seriously, the authors seem not to understand statistics. Repeatedly, they state expression differences, followed by the statement that results were not significant. If a difference fails the test of significance that means there is NO difference in expression. This needs to be stated clearly and without equivocation

In addition, of their 7 candidate microRNAs, 5 failed to be validated by qPCR, meaning that their initial observation was spurious.

The authors do not discuss the frequent high variance in expression among samples within a group. They need additional analyses to verify that their results are not due to outliers. For example, in Figure 1A for mir135, ½ the AVF sample values are clustered at the same level as normals, and the other half are highly variable. If outliers are deleted, likely the significant difference would disappear. The same problem is present in the decrease of mir137.

The high variability means that the authors needs many more samples. It is also possible that the high variability is due to collection times and processing delays with surgerical samples.

2. Why are data presented for only 6 and 8 samples in Figure 2 etc, instead of for the original 10 and 13? All samples must be analyzed and results presented. This is a serious deficit

3. In several panels of each figure, data are duplicated without clearly stating that fact. Graphical presentation could be morely clearly presented with all conditions in one graph, e.g normal untreated, normal + hypoxia, normal + VEGF, normal + hypoxia and VEGF, and AVF untreated, AVF + hypoxia etc. Then there is no duplication of data and all comparisons are together. (except it is not easily clear that some experiments have not been repeated??)

Comments on the Quality of English Language

4. The manuscript needs extensive editing throughout. One example: paragrpah 2.1.1 Use either present or past tense (usually, past tense is used). Do not reproduce a lab protocol format (i.e. use complete sentences, not “bullet points”).

Correct the section numbers in the Methods

Change erroneous capital letters (e.g. line 199 Hypoxia)

In general, I doubt p values are accurate to 4 significant figures, even if the program provides it.

Delete mentions of “VEGF-induced angiogenesis” because angiogenesis has not been assessed

Author Response

  1. Most seriously, the authors seem not to understand statistics. Repeatedly, they state expression differences, followed by the statement that results were not significant. If a difference fails the test of significance that means there is NO difference in expression. This needs to be stated clearly and without equivocation

> Yes, your mentioned comment is accurate. While it was not statistically significant, we highlighted the observed trend in differences. We deemed understanding such trends crucial in the identification of novel microRNAs. Even though statistically insignificant, recognizing such trends is highly important in the process of discovering novel microRNAs

In addition, of their 7 candidate microRNAs, 5 failed to be validated by qPCR, meaning that their initial observation was spurious.

> We conducted profiling on seven entities, and the validation process was carried out to confirm the results. It is true that five of them did not undergo validation. However, in the profiling process aimed at identifying novel microRNAs and selecting those with potential relevance to AVM, it does not imply that all profiles must pass validation. We have undertaken comprehensive revisions throughout the manuscript, and we kindly request a positive review. Thank you for your keen comment.

The authors do not discuss the frequent high variance in expression among samples within a group. They need additional analyses to verify that their results are not due to outliers. For example, in Figure 1A for mir135, ½ the AVF sample values are clustered at the same level as normals, and the other half are highly variable. If outliers are deleted, likely the significant difference would disappear. The same problem is present in the decrease of mir137.

> We apologize for making confusion. In order to facilitate a meaningful comparison of data values within the same graph, the y-axis of the graph was normalized to RNU6B.

Those spots you perceived as outliers in the background were intended to highlight the changes on the graph due to the small unit scale in between. Therefore, there shouldn't be a significant difference here, as the purpose was to visualize the variation Consequently, there is a range of 0.05 to 25 in the maximum values of the between the graphs. We clarified it under the legends. Thank you for bringing it to my attention.

The high variability means that the authors needs many more samples. It is also possible that the high variability is due to collection times and processing delays with surgical samples.

> Thank you for your valuable comments. The majority of the samples were collected and processed by the planned schedule, usually within the same timeframe. While we generally considered minimal variation due to this consistency, we acknowledge the possibility of slight differences. In future studies, we aim to address these considerations more precisely as you mentioned.

Moreover, the rarity of the disease posed challenges in accumulating a large number of samples. We will continue our research efforts to gradually increase the sample size.

  1. Why are data presented for only 6 and 8 samples in Figure 2 etc, instead of for the original 10 and 13? All samples must be analyzed and results presented. This is a serious deficit

> Ten and thirteen findings through profiling is accurate. Experiments in section 2.2 were conducted with various conditions after profiling. Due to the need for a substantial number of samples for this phase, a significant amount was required. Unfortunately, during storage, there was an unexpected malfunction in the storage tank device, resulting in a slight decrease in the actual number of samples for experimentation. This sample number truthfully reflects this unfortunate circumstance.

We regret this inconvenience and are actively working to minimize such disruptions in the future by implementing careful management of equipment and machinery. We appreciate your comments. The manuscript has revised to clear confusion on line 162-163 and Table 1.

  1. In several panels of each figure, data are duplicated without clearly stating that fact. Graphical presentation could be morely clearly presented with all conditions in one graph, e.g normal untreated, normal + hypoxia, normal + VEGF, normal + hypoxia and VEGF, and AVF untreated, AVF + hypoxia etc. Then there is no duplication of data and all comparisons are together. (except it is not easily clear that some experiments have not been repeated??)

> We admit your considerable comment. However, as we explained above, there is a wide range from 0.05 to 25 in the maximum values of each y-axis, so I think it would be more effective to represent the graphs separately for each microRNA. Thank you for your attention to detail.

  1. The manuscript needs extensive editing throughout. One example: paragrpah 2.1.1 Use either present or past tense (usually, past tense is used). Do not reproduce a lab protocol format (i.e. use complete sentences, not “bullet points”).

Correct the section numbers in the Methods

Change erroneous capital letters (e.g. line 199 Hypoxia)

In general, I doubt p values are accurate to 4 significant figures, even if the program provides it.

Delete mentions of “VEGF-induced angiogenesis” because angiogenesis has not been assessed

>Thank you for your insightful comments. We have undergone professional editing from a certified manuscript editing service. Following your feedback, we have made overall revisions, addressing issues such as capitalization errors, grammar, and the deletion of lab protocol formats. We have also revised the mention  'VEGF-induced angiogenesis' (line 251, 335) as you suggested

Reviewer 3 Report

Comments and Suggestions for Authors

The manuscript by Lee, et al. shows an interesting finding that miR-135b-p5 is upregulated in endothelial cells isolated from AVM patients when compared to normal controls. However, this reviewer found the methods and results to be overly simplified and observational, which reduced my enthusiasm for this manuscript. I therefore believe this study is not fit for the current Journal. Please find my specific concerns below:

Major Comments:

1) There is a lack of mechanistic evaluation in the current study making it difficult to know if miR-135b-5p is actively pathogenic in AVM, or if it is only an association. Therefore, I would recommend a major series of experiments that include miR-135b-5p loss or gain-of-function studies, as well as functional experiments to measure EC angiogenesis, proliferation, barrier function, dysfunction, etc.

2) VEGF and hypoxia have many functional pathways in endothelial cells. Therefore, it is an over simplification to state VEGF is angiogenesis, for example.

3) The title states “upregulation of miR-135b-p5 promotes pathophysiological processing in AVM ECs”. However, this is an overstatement as functional studies must be included in order to state any pathophysiological relevance of miR-135b-p5, and there were no gain-of-function studies regarding miR-135b-5p.

4) Figure 1 indicates significant increase in miR-135b-5p in AVM endothelial cells. However, Figure 2 does not show this increase in normal vs AVM ECs when under normoxia. The increase in miR-135b-5p is only shown with hypoxia or VEGF treatment. Can this be explained?

5) Do you think ECs collected from AVM patients, but at normal sites of vasculature, would also have increased miR-135b-5p? Knowing this is difficult to collect from the same patient, this can just be commented upon.

6) A table of patient characteristics must be included to list patient age, sex, comorbidities, etc.

7) Why weren’t all cell lines used for the qPCR experiments. If there were 10 normal and 13 AVM EC cell lines available, why were only 6 vs 8 cell lines, respectively, used in the qPCR evaluation?

8) HIF1a was mentioned a lot in the discussion. What are the HIF1a measurements in normal vs AVM endothelial cells?

9) The figure layout and text make this manuscript quite complicated to read. I would recommend changing the labeling of the figures to A-J, instead of A1, A2, A3. Furthermore, referring to the treatment groups in the text is quite hard to understand and causes the reader to go back and forth from the methods section to figure out which treatment the authors are referring to. I would remove the grouping and clearly state the treatments in the text.

10) Figure legends should only contain information regarding the experimentation and statistical analysis, no results should be listed here.

Comments on the Quality of English Language

I would suggest editing the manuscript by a native english speaker.

Author Response

1) There is a lack of mechanistic evaluation in the current study making it difficult to know if miR-135b-5p is actively pathogenic in AVM, or if it is only an association. Therefore, I would recommend a major series of experiments that include miR-135b-5p loss or gain-of-function studies, as well as functional experiments to measure EC angiogenesis, proliferation, barrier function, dysfunction, etc.

3) The title states “upregulation of miR-135b-p5 promotes pathophysiological processing in AVM ECs”. However, this is an overstatement as functional studies must be included in order to state any pathophysiological relevance of miR-135b-p5, and there were no gain-of-function studies regarding miR-135b-5p.

> 1),3) 

Thank you for the valuable input. Our study focuses on up and down regulation of miRNA and significant changes in endothelial cells of AVM under conditions that vary with the presence of hypoxia and VEGF treatment, which conditions previously shown to have angiogenesis effects. We acknowledge the limitation highlighted, as we did not conduct experiments under conditions of loss or gain of function for miR-135b-5p.

2) VEGF and hypoxia have many functional pathways in endothelial cells. Therefore, it is an over simplification to state VEGF is angiogenesis, for example.

> Thank you for the sharp comment. Though we were focusing on the angiogenesis effect, we admit It was indeed a stretch to solely characterize the effects of VEGF as angiogenesis. Following your advice, We have added references to provide a more nuanced and accurate representation regarding VEGF. (line 73-77)

4) Figure 1 indicates significant increase in miR-135b-5p in AVM endothelial cells. However, Figure 2 does not show this increase in normal vs AVM ECs when under normoxia. The increase in miR-135b-5p is only shown with hypoxia or VEGF treatment. Can this be explained?

> Thank you for the insightful feedback. We have incorporated the points you raised into the limitations section, on line 367

5) Do you think ECs collected from AVM patients, but at normal sites of vasculature, would also have increased miR-135b-5p? Knowing this is difficult to collect from the same patient, this can just be commented upon.

> Thank you for your comment. We are actively continuing additional research on the aspects you mentioned. It would have been ideal to experiment with a sufficient number of endothelial samples from vessels where AVM did not occur in AVM patients, however, ethical problems made it challenging to collect normal tissues from areas unrelated to AVM treatment surgeries. Although, in some cases involving flap surgery, attempts were made to collect samples, but still having difficulty in obtaining an enough samples.

Therefore, we are planning to explore the analysis of circulating RNA and blood samples in future research. If successful, we will certainly provide your journal additional reports on this aspect. While it remains to be confirmed through further analysis, an increase in the miR-135b-5p is anticipated compared to normal patients. We sincerely appreciate the thorough review.

6) A table of patient characteristics must be included to list patient age, sex, comorbidities, etc.

> We have added the sample characteristic table (Table 1). We have excluded the patients with comorbidities other than AVM.

7) Why weren’t all cell lines used for the qPCR experiments. If there were 10 normal and 13 AVM EC cell lines available, why were only 6 vs 8 cell lines, respectively, used in the qPCR evaluation?

> Experiments detailed in section 2.2 were conducted under various conditions subsequent to the profiling. Substantial samples were needed on this phase. Unfortunately, There was a breakdown on storage tank device and several samples were disposed. The reported sample number truthfully reflects this circumstance.

We appreciate your comments and will minimize loss on future experiment.

8) HIF1a was mentioned a lot in the discussion. What are the HIF1a measurements in normal vs AVM endothelial cells?

> Your comments are greatly appreciated, and I consider them to be very insightful. It seems to be a promising topic for exploration in our future research. Thank you

9) The figure layout and text make this manuscript quite complicated to read. I would recommend changing the labeling of the figures to A-J, instead of A1, A2, A3. Furthermore, referring to the treatment groups in the text is quite hard to understand and causes the reader to go back and forth from the methods section to figure out which treatment the authors are referring to. I would remove the grouping and clearly state the treatments in the text.

> Thank you. I acknowledge the concern about the alphabetic listing in the expression, and we have made it clear in the text and also revised the grouping in the figures accordingly.

10) Figure legends should only contain information regarding the experimentation and statistical analysis, no results should be listed here.

Thank you, we have deleted the result on every figure legends in the manuscript.

Reviewer 4 Report

Comments and Suggestions for Authors

Dear Editor,

I read with great interest this paper from Lee et al. 

The manuscript dealt with the role of microRNA 135b-5p on the endothelium of arteriovenous malformations. The subject is really interesting due to the rarity of the diseases and the lack of information about the management.

I would only suggest to include more numerical data in the results section of the abstract. 

The English of the paper is fluent and the paper is comprehensive.

The scientific background is solid.

Author Response

Dear Editor,

I read with great interest this paper from Lee et al. 

The manuscript dealt with the role of microRNA 135b-5p on the endothelium of arteriovenous malformations. The subject is really interesting due to the rarity of the diseases and the lack of information about the management.

I would only suggest to include more numerical data in the results section of the abstract. 

The English of the paper is fluent and the paper is comprehensive.

The scientific background is solid.

> Your generous complements make us feel rewarded in our research, I appreciate it. We apologize for the oversight; We were rather focused on elucidating which miRNA had significant implications under what conditions in the results of this paper that we overlooked clear numerical data. We have addressed your comment by adding the p-values indicating statistical significance in the abstract's results section. We will take a step further and present clearer results through numerical data in the following research.

Round 2

Reviewer 1 Report

Comments and Suggestions for Authors

I congratulate with the authors for making all the required revisions. No further comments. The article is suitable for publication.

Author Response

Thank you for your consideration.

Reviewer 3 Report

Comments and Suggestions for Authors

The authors have commented on some of my original concerns. However, there must be additional experiments made to improve the quality of this manuscript in order to fit the journal. Furthermore, many of the comments do not adequately address my concerns.

Comments on the Quality of English Language

Manuscript should be edited by a native english speaker.

Author Response

The authors have commented on some of my original concerns. However, there must be additional experiments made to improve the quality of this manuscript in order to fit the journal. Furthermore, many of the comments do not adequately address my concerns.

A) We apologize for providing an inadequate answer by not properly focusing on your comment. We have prepared a response again and kindly request your positive review. Thank you once again for your meticulous review.

Comments on the Quality of English Language

Manuscript should be edited by a native english speaker.

A) We have resubmitted for formal English correction. The deficiencies have been revised. Thank you.

1) There is a lack of mechanistic evaluation in the current study making it difficult to know if miR-135b-5p is actively pathogenic in AVM, or if it is only an association. Therefore, I would recommend a major series of experiments that include miR-135b-5p loss or gain-of-function studies, as well as functional experiments to measure EC angiogenesis, proliferation, barrier function, dysfunction, etc.

A) In this report, we focused on selecting microRNAs in AVM that seemed relevant to angiogenesis and proceeded with experiments accordingly. In our study, we established conditions based on hypoxia and VEGF treatment, and statistically analyzed the resulting changes. As you mentioned, we are currently preparing to continuously perform functional studies in subsequent research. Your comments have helped us to refine our direction. Thank you. We will prepare to present more improved research in our future studies. Thank you again.

2) VEGF and hypoxia have many functional pathways in endothelial cells. Therefore, it is an over simplification to state VEGF is angiogenesis, for example.

A) In the process of adjusting the flow of the manuscript, it seems that the part about VEGF was oversimplified. Therefore, we have added references to offer a more nuanced and accurate depiction regarding VEGF (line 73-78 / line 328-332 / line 342-349). We kindly request your positive review once again. Thank you.

3) The title states “upregulation of miR-135b-p5 promotes pathophysiological processing in AVM ECs”. However, this is an overstatement as functional studies must be included in order to state any pathophysiological relevance of miR-135b-p5, and there were no gain-of-function studies regarding miR-135b-5p.

A) As you pointed out, and as mentioned in our first answer, we agree that the title of our study was overly stated, considering it primarily analyzed changes based on conditions alone. Therefore, we have revised the title to 'MicroRNA-135b-5p expression as a promising biomarker in the endothelial cells of arteriovenous malformations'. We apologize for initially choosing an exaggerated title and are grateful for your sharp comments, which helped clarify potential confusion in the title. We kindly request your positive review on this revision. Thank you

4) Figure 1 indicates significant increase in miR-135b-5p in AVM endothelial cells. However, Figure 2 does not show this increase in normal vs AVM ECs when under normoxia. The increase in miR-135b-5p is only shown with hypoxia or VEGF treatment. Can this be explained?

A) We also discussed the statistical aspects when composing each figure. It seems that the relevant part was omitted during the image work for Figure 2. We have now amended Figure 2 to reflect statistical significance. We apologize for any confusion caused. Initially, we misunderstood your review. Sorry for that.

5) Do you think ECs collected from AVM patients, but at normal sites of vasculature, would also have increased miR-135b-5p? Knowing this is difficult to collect from the same patient, this can just be commented upon.

Our response to this comment remains the same as before, but if you could specify what aspects were significantly lacking, we will strive to address them further. We share the same perspective as you, the reviewer. In future research, once confirmed, we plan to incorporate these findings into the manuscript. Thank you.

(review 1st stage answers) Thank you for your comment. We are actively continuing additional research on the aspects you mentioned. It would have been ideal to experiment with a sufficient number of endothelial samples from vessels where AVM did not occur in AVM patients, however, ethical problems made it challenging to collect normal tissues from areas unrelated to AVM treatment surgeries. Although, in some cases involving flap surgery, attempts were made to collect samples, but still having difficulty in obtaining an enough samples.

Therefore, we are planning to explore the analysis of circulating RNA and blood samples in future research. If successful, we will certainly provide your journal additional reports on this aspect. While it remains to be confirmed through further analysis, an increase in the miR-135b-5p is anticipated compared to normal patients. We sincerely appreciate the thorough review.

6) A table of patient characteristics must be included to list patient age, sex, comorbidities, etc.

A) We have added the patient characteristic table (Table 1) on line 136 including age, sex, comorbidities. We have excluded the patients with comorbidities other than AVM. As you suggested, organizing the patient characteristics into a table has indeed made the paper more concise and clear for other readers, resulting in a more focused and engaging manuscript. Thank you.

7) Why weren’t all cell lines used for the qPCR experiments. If there were 10 normal and 13 AVM EC cell lines available, why were only 6 vs 8 cell lines, respectively, used in the qPCR evaluation?

A) Our team always believes that in experimental clinical studies, it is essential to convey facts and share the results with others conducting similar research. As you mentioned, we were preparing to conduct the experiment with consistent numbers, but due to issues with the sample storage device, the number of samples slightly decreased. This reduction has been accurately noted. Any potential confusion was clarified by presenting this information in the patient characteristics table. We are very grateful for your comment on this aspect. We kindly request your positive review. Thank you.

8) HIF1a was mentioned a lot in the discussion. What are the HIF1a measurements in normal vs AVM endothelial cells?

A) We apologize for the inadequate response to the comment in the first review. Currently, there seems to be no reference comparing endothelial cells in normal and AVM groups, possibly due to the rarity of AVM as a disease. The reason for focusing on HIF-1α in our manuscript originated from a strong conjecture that the following process could be related to AVM.

We have added additional reference regarding HIF1-alpha on line 315-322 and 328-332. Under normoxic conditions, HIF-1α, a subunit of the oxygen-sensitive transcription factor HIF-1, is rapidly degraded by a process involving prolyl hydroxylases and the von Hippel-Lindau ubiquitin E3 ligase complex. This regulation of HIF-1 plays a crucial role in gene activation related to angiogenesis in response to varying oxygen levels. And under hypoxic conditions, the HIF-1α subunit is stabilized and translocates to the nucleus to form a complex with HIF-1β, thereby activating genes crucial for angiogenesis and red blood cell production, including those encoding VEGF-A and erythropoietin (EPO).”

Thank you for your comment. We believe that focusing on your comment in our next study could bring us closer to a more accurate understanding of the pathophysiology of endothelial cells in AVM.

9) The figure layout and text make this manuscript quite complicated to read. I would recommend changing the labeling of the figures to A-J, instead of A1, A2, A3. Furthermore, referring to the treatment groups in the text is quite hard to understand and causes the reader to go back and forth from the methods section to figure out which treatment the authors are referring to. I would remove the grouping and clearly state the treatments in the text.

A) Thank you. I acknowledge the concern about the alphabetic listing in the expression, and we have made it clear in the text and also revised the grouping in the figure 2 and 3 accordingly on line 230 and 246 each.

The graphs depicting each RNA type have different y-axis scales, which means that even similar differences on the graph represent varying magnitudes of change. This approach was chosen to enhance visibility and clarify even small variations in each RNA type. We kindly request your positive review of this aspect. Thank you.

10) Figure legends should only contain information regarding the experimentation and statistical analysis, no results should be listed here.

A) you suggested, we have removed the experimental results that were redundantly mentioned in the figure legend and the results section- line 232-236 on figure 2 and line 247-249 on figure 3.
